# The Impact of Quantifying Human Locomotor Activity on Examining Sleep–Wake Cycles

**DOI:** 10.3390/s25247659

**Published:** 2025-12-17

**Authors:** Bálint Maczák, Adél Zita Hordós, Gergely Vadai

**Affiliations:** Department of Technical Informatics, University of Szeged, 6720 Szeged, Hungary; adelhordos@gmail.com (A.Z.H.); vadaig@inf.u-szeged.hu (G.V.)

**Keywords:** actigraphy, human locomotor activity, sleep–wake cycle, circadian rhythm

## Abstract

Actigraphy quantifies human locomotor activity by measuring wrist acceleration via wearable devices at relatively high rates and converting it into lower-temporal-resolution activity values; however, the computational implementations of this data compression differ substantially across manufacturers. Building on our previous work comparing activity determination methods, we have investigated how they (e.g., digital filtering and data compression) influence nonparametric circadian rhythm analysis and sleep–wake scoring. In addition to our generalized actigraphic framework, we have also emulated the use of specific devices commonly employed in such sleep-related studies by applying their methods to raw actigraphic acceleration data we collected to demonstrate, through concrete real-life examples, how methodological choices may shape analytical outcomes. Additionally, we assessed whether nonparametric indicators could be derived directly from acceleration data without compressing them into activity values. Overall, our analysis revealed that all these analytical approaches to the sleep–wake cycle can be substantially affected by the manufacturer-dependent actigraphic methodology employed, with the observed effects traceable to distinct steps of the signal-processing pipeline, underscoring the necessity of cross-manufacturer harmonization from a clinically oriented perspective.

## 1. Introduction

Actigraphy has been widely used since the 1980s to quantify human motor activity [1]. The method employs a research-grade wearable accelerometer, typically worn on the non-dominant wrist, which samples acceleration mostly along three axes at relatively high rates (e.g., 10–100 Hz). These samples are subsequently aggregated into fixed-length epochs (e.g., 1 min) by the actigraph device to yield activity values stored in on-board memory [2]. The derivation of these activity values constitutes a lossy data reduction step implemented through manufacturer-specific and largely proprietary algorithms that obscure the underlying methods from end users [3]. Although such lossy compression was historically necessitated by limited memory, contemporary devices can also store the raw, high-temporal-resolution acceleration data. However, when raw data (hereafter referred to as acceleration data) are recorded, one may need to transform them post hoc into activity values (hereafter referred to as activity data) to align with their specific analytic approaches, yet no standardized pathway exists due to the historical heterogeneity of classical devices. In our previous work [2], we collected and categorized the various activity determination methods prevalent in the literature and compared them by applying them to the same set of acceleration data. Through correlation-based analysis, we found that these methods can quantify the same locomotor activity substantially differently, which complicates reproducibility and cross-study comparability. Elements of the generalized activity determination framework and nomenclature we introduced in that work have reportedly been adopted by Garmin Ltd. (Olathe, KS, USA), a market-leading sports watch manufacturer [4], underscoring the practical relevance of cross-manufacturer harmonization.

Despite these challenges, actigraphy is widely applied across disciplines, including medical and psychiatric research [5,6], sports science [7], and analysis of daily motion patterns [8]. In sleep research, actigraphy provides less physiological detail than the gold-standard laboratory polysomnography (PSG) approach but offers the major advantage of noninvasive, continuous monitoring over weeks in everyday settings. Nonparametric circadian rhythm analysis (NPCRA) is frequently applied to the collected activity data to characterize circadian and rest–activity rhythms in subjects with various health conditions [9], with commonly derived indicators including the onset time and mean activity of the least active 5 (L5) and most active 10 (M10) consecutive hours. Another common application is sleep–wake scoring based on actigraphic data, for which numerous algorithms have been proposed; classic scoring algorithms (e.g., the Sadeh [10] and Cole–Kripke [11] algorithms) operate solely on epoch-level activity data, whereas the availability of raw acceleration recordings from modern devices has enabled acceleration–based approaches (e.g., the van Hees algorithm [12]).

However, considering our previous comparative work [2], in which we underscored the discrepancies between activity determination methods, the question arises as to how strongly the results of NPCRA and sleep–wake scoring depend on manufacturer-specific approaches to locomotor activity quantification and, pragmatically speaking, on device selection. In the context of NPCRA, this question has not yet been systematically examined, even though studies frequently employ devices from different manufacturers—e.g., Ametris LLC (Pensacola, FL, USA, formerly ActiGraph LLC) [13,14,15], CamNtech Ltd. (Cambridge, UK) [16,17], Condor Instruments (São Paulo, Brazil) [18,19,20,21], Ambulatory Monitoring Inc. (Ardsley, NY, USA) [21,22,23,24,25]—and thus analyze activity signals produced by heterogeneous processing pipelines (see Section 2.2.2 for details), directly compromising cross-study comparability. For sleep–wake scoring, only one work has evaluated how results obtained with the Sadeh algorithm vary with the consumer wearable used to record raw acceleration; however, the raw data were compressed to epoch-level form in only a single way using the Actilife software product developed by Ametris, leaving the broader landscape of device-specific effects unmapped [26].

To address these medical-application-oriented questions, as shown in Figure 1, we calculated manufacturer-specific activity data based on already-collected raw actigraphic acceleration data to retrospectively emulate different devices. Then, we derived NPCRA indicators and scored sleep based on all these activity data, which quantify the same motion in alternative ways. Finally, we compared the circadian-rhythm-related analytical outcomes (i.e., the features we extracted) to assess the extent to which device selection influences them. Beyond the medical relevance of our examination, it can also serve as a continuation of our earlier correlation-based comparison of activity determination methods. Here, we also examine how NPCRA indicators behave when computed directly based on acceleration data rather than activity data. This unprecedented approach allows us to evaluate the similarities between high-temporal-resolution acceleration signals and epoch-level activity data beyond comparing their spectral properties [27]—an examination that cannot be performed with correlation analysis due to the differing temporal resolutions of the two signal types.

To reveal the differences caused by the heterogeneity of actigraphic methodologies in circadian rhythm analysis, we break down the depicted steps (Figure 1) in reverse order in Section 2 (Materials and Methods). After introducing the sleep-related features we extracted via NPCRA and sleep–wake scoring (Section 2.1), we define the different types of actigraphic signals—each characterizing the same motion—from which these features are derived and explain how we calculated these signals (Section 2.2) based on the raw acceleration recordings (Section 2.3.1). Finally, we outline the procedure used to compare the medical features derived from differently defined activity data (Section 2.3.2). In Section 3 (Results), we present the results of our analysis, first for the generalized activity determination methods (Section 3.1) and then for those corresponding to specific commercially available devices (Section 3.2). Lastly, Section 4 (Conclusions and Discussion) summarizes our main findings and discusses their implications and limitations.

## 2. Materials and Methods

### 2.1. Extracting Sleep-Related Features

#### 2.1.1. Sleep–Wake Scoring

Over the past four decades, numerous heuristic sleep–wake scoring algorithms have been developed, with more recent approaches incorporating machine learning [1]. In this study, we used one representative algorithm designed for epoch-level activity data and another for high-temporal-resolution acceleration data to assess the impact of actigraphic signal processing on sleep–wake classification, aiming to detect sustained, consolidated sleep periods for the sake of comparability.

The Munich Actimetry Sleep Detection Algorithm (MASDA, also known as Roenneberg algorithm) is a device-independent activity-data-based sleep–wake scoring algorithm that was validated on ActTrust devices produced by Condor Instruments and subsequently applied to activity data collected with MotionWatch 8 devices developed by CamNtech Ltd. [28,29]. Briefly, MASDA estimates the trend of activity data using a centered 24 h moving average and classifies an epoch as sleep if its activity value falls below 15–25% of the trend; the resulting raw sleep–wake classification then undergoes two-stage automatic rescoring, yielding long stretches of consolidated sleep and wake [30]. For this analysis, the open-source pyActigraphy (v1.2.1) implementation [31] of MASDA was used with default parameters, and the threshold was set to 25% as per the recommendation [28].

The van Hees algorithm from GGIR (a research-community-driven R package developed for transparent, vendor-agnostic processing and analysis of actigraphic acceleration data) is specially designed for acceleration-data-based sleep–wake scoring. Briefly, it relies on two parallel steps whose combination yields the final classification [32]: the detection of sustained inactivity bouts (SIBs [12]) and sleep period time (SPT [33]) window. SIBs are identified when the change in wrist angle remains within 5° for at least 5 min. SPT-window detection also relies on wrist angle but includes an automated cleaning step to ensure there is a single contiguous SPT per day (noon-to-noon). Sleep is defined based on the overlap between the contiguous SPT and multiple shorter SIBs. In this study, GGIR (v3.2.6) was run with default settings, except that the SPT threshold was set to 0.4 following visual inspection; which is permitted for methodological research according to the creators of the package [34]. Note that since the acceleration data we have examined (sampled at 10 Hz—see Section 2.3.1) included real-time clock-based, millisecond precision time stamps, it was necessary to reduce the resolution of the time stamps to obtain time-equidistant samples spaced exactly 100 milliseconds apart, as required by the algorithm.

#### 2.1.2. Nonparametric Measures of Circadian Rhythm

The characterization of biological rest–activity rhythms has a long history. In this task, features are extracted from activity data to quantify distinct aspects of the rhythm. Here, the effects of actigraphic signal processing were assessed for the 5 most-used indicators in nonparametric circadian rhythm analysis [9].

L5 and M10 capture the least-active 5 h and most-active 10 h windows within a day with onset times (L5onset and M10onset) and mean activity levels (L5val and M10val) [9]. Thus, while L5 indexes nocturnal activity, M10 indexes diurnal activity: a lower L5val implies more restful sleep, and a higher M10val indicates greater daytime activity. Relative Amplitude (RA) is used to compute the normalized difference between M10val and L5val (see Equation (1)). RA ranges from 0 to 1; larger values indicate greater separation between nighttime and daytime activity and are generally taken to reflect a more robust rest–activity rhythm. We determined L5 and M10 in the traditional way [13,14,15,17,18,19,20,21,22,23,24,35,36]: first, a 24 h mean profile was obtained by averaging the minute-level activity data across days; then, L5 and M10 were derived by sliding a window in one-minute steps across the 24 h mean profile in a circular manner and computing the mean activity within each window.(1)$$RA=\frac{M10_{val}-L5_{val}}{M10_{val}+L5_{val}}$$

Interdaily Stability (IS) quantifies the day-to-day predictability (i.e., similarity) of a subject’s activity pattern and is computed using Equation (2), where $x_{i}$ denotes the mean activity in each hour in the multi-day activity data, $x_{m}$ is the grand mean across the entire time series, and $x_{h}$ is the mean activity of the h-th hour in the 24 h mean profile [9]. IS ranges from 0 to 1; larger values denote more consistent day-to-day activity patterns.(2)$$IS=\frac{\frac{1}{24}\sum_{h=1}^{24}{\left(x_{h}-x_{m}\right)}^{2}}{\frac{1}{N}\sum_{i=1}^{N}{\left(x_{i}-x_{m}\right)}^{2}}$$

Intradaily Variability (IV) measures within-day fragmentation of the rest–activity rhythm [9] and is computed as shown in Equation (3), where $x_{i}$ denotes the mean activity in each hour in the multi-day activity data, and $x_{m}$ is the grand mean across the entire time series. IV ranges from 0 to 2; higher values indicate more frequent rest–activity transitions (e.g., daytime naps and nighttime awakenings), i.e., greater fragmentation.(3)$$IV=\frac{\frac{1}{N-1}\sum_{i=2}^{N}{\left(x_{i}-x_{i-1}\right)}^{2}}{\frac{1}{N}\sum_{i=1}^{N}{\left(x_{i}-x_{m}\right)}^{2}}$$

### 2.2. Actigraphic Signal-Processing Pipelines

Both NPCRA indicators and the MASDA sleep–wake scoring algorithm defined in the previous sections were applied to epoch-level activity data, but to assess the effect actigraphic signal processing methods have on them, we first need to understand how activity data can be derived from the raw acceleration data how these activity determination methods differ between devices. In Section 2.2.1, we introduce our generalized activity determination framework building on our prior work [2], where we systematically collected, categorized, and compared activity determination methods and abstracted them into the following two-step processing pipeline: (1) preprocessing of raw acceleration data and (2) activity-metric-based aggregation to epoch-level values. Although individual operations in our generalized framework map to commercially available solutions, the entire activity determination workflows—specific pairings of preprocessing techniques and activity metrics—are difficult to match one-to-one to any specific device. Therefore, to provide a comprehensive yet realistic assessment of how signal-processing steps affect NPCRA indicators and sleep–wake scoring, we not only rely on our generalized framework (see Section 2.2.1) to determine activity values from the same set of wrist motion data (see Section 2.3.1) but also extend our analysis to methods employed in specific devices (see Section 2.2.2) that are frequently used in the related literature.

#### 2.2.1. Generalized Activity Determination Methods

In most cases, actigraphs using MEMS (micro-electromechanical system) accelerometers measure the x-, y-, and z-axis components of limb acceleration (UFX, UFY, and UFZ, collectively termed UFXYZ). During preprocessing, the primary aims are to combine these axial signals into a single series and remove the effect of Earth’s gravitational acceleration (i.e., the gravitational component, *g*). The former can be achieved by computing the Euclidean norm of the per-axis accelerations to obtain the unfiltered magnitude data (UFM). The effect of *g* can be removed from this magnitude data by subtracting 1 g and either taking the absolute value (UFNM) or truncating negative differences to zero (ENMO). Alternatively, the DC (gravitational) component can be attenuated using digital filters (e.g., a 3rd-order Butterworth band-pass filter with 0.25 and 2.5 Hz cutoffs), applied either to the UFM data (FMpost) or to the per-axis signals (FX, FY, and FZ, collectively denoted as FXYZ) before computing the Euclidean norm (FMpre).

Following preprocessing, the acceleration data is partitioned into fixed-length epochs (in our case, 1 min), and an activity metric is applied to yield one activity value per epoch. The most classical metrics either simply integrate the acceleration (Proportional Integration Method—PIM) or use fixed thresholds to determine the crossings (Zero Crossing Method—ZCM) or the time the signal remains above such thresholds (Time Above Threshold—TAT). Contrary to the name, ZCM, the threshold level of these intersection-based activity metrics should be set above the noise level rather than at zero, though its exact value has not been disclosed by the manufacturers. Previously [2], we determined its optimal level, and we have accordingly set it equal to the standard deviation (note that for UFM, 1 g should be added to account for the presence of the gravitational component) of all the preprocessed acceleration data in this analysis. Other metrics are based on calculating the standard deviation (Mean Amplitude Deviation—MAD) or the variance (Activity Index—AI) of the acceleration data in a given epoch. The metric High-Pass Filtered Euclidean Norm (HFEN) simply averages the acceleration data, but it requires a specific preprocessing scheme: the per-axis signals must be filtered with a high-pass—not a band-pass—filter (4th-order Butterworth high-pass filter with 0.2 Hz cutoff) before magnitude calculation (denoted as HFMpre rather than FMpre).

Preprocessing techniques and activity metrics can be properly combined in 35 different ways, each of which is a different activity determination method that we denote here in accordance with our past work: the activity metric is treated as a function whose argument is the preprocessing technique. Thus, for example, PIM(UFNM) denotes such an activity signal that has been obtained by minutely integrating the normalized magnitude of acceleration. For more details of these procedures, see our prior work [2].

#### 2.2.2. Activity Determination Methods for Specific Devices

Given the widespread use of devices from Ametris LLC [13,14,15] and CamNtech Ltd. [16,17] in NPCRA studies, the corresponding activity determination methods have also been included in the present analysis. We have used the Python implementation of Ametris’s algorithm (agcounts, v0.6.9), which the company published 3 years ago to enhance transparency [37] to address the scientific community’s demand for accessible and reproducible methods [3]. Their solution is comparatively complex and does not align one-to-one with the methods in Section 2.2.1, as epoch-level activity values (the so-called Activity Count; hereafter referred to as AC) are computed via multiple stages (resampling, band-pass filtering, rescaling, thresholding, and summing—see [3] for details). CamNtech disclosed the activity determination procedure of their MotionWatch 8 devices in their user manual [38]: each axis is band-pass-filtered prior to magnitude computation (FMpre-like preprocessing, with 3 and 11 Hz cutoffs); per-second peak acceleration is then determined, and peaks exceeding 0.1 g are summed within each epoch to determine the activity value (hereafter referred to as MW). As the acceleration data we examined were recorded at a sampling rate of 10 Hz, we have approximated this band-pass filter using the one specified in our generalized framework.

Beyond these approaches, the PIM(ENMO) activity-determination method in our generalized scheme corresponds directly to the widely used GGIR package (v3.2.6), which applies ENMO by default and derives epoch-level values through simple averaging, although it uses 5 s epochs by default [32]. However, since correlation-based examinations require consistent epoch length across activity determination methods and in circadian rhythm analysis—where the focus is on the overall multi-day activity pattern rather than short bursts—a 1 min epoch length is preponderant and recommended [13,16,17,18,20,23,39] and the use of ENMO with 1 min epochs is not unprecedented [40], we employed it as well. Furthermore, the ZCM(FMpre), TAT(FMpre), and PIM(FMpre) methods correspond to those implemented in Condor Instruments’ ActTrust devices [18,19,20,21], with the sole difference being that ActTrust employs a 0.5–2.7 Hz band-pass filter rather than the 0.25–2.5 Hz band-pass used here [41]. Note that the ZCM, TAT, and PIM metrics are also supported by long-standing and widely used devices produced by Ambulatory Monitoring Inc. [21,22,23,24,25]; however, because these devices use piezoelectric rather than MEMS sensors and only fragmented information is available regarding their algorithmic details [42], the correspondence is necessarily indirect.

### 2.3. Comparison of Signal-Processing Pipelines Through Extracted Features

In our analysis, we utilized a dataset of raw actigraphic acceleration data, from which we were retrospectively able to calculate various epoch-level activity data (and higher-temporal-resolution acceleration data preprocessed in different ways, as a preliminary step) using the methods presented in Section 2.2 to emulate the activity determination approaches of different manufacturers. Subsequently, we applied NPCRA and sleep–wake scoring algorithms to these differently calculated signals to assess the disparities introduced in their results by the choice of locomotor activity quantification method.

#### 2.3.1. Acceleration Dataset

The publicly available actigraphic dataset we collected and examined in several of our previous studies [2,6,27], as well as in this work, contains raw triaxial acceleration data obtained from 42 healthy, free-living subjects [43]. The 10-day-long recordings were acquired on each subject’s non-dominant wrist at a sampling rate of 10 Hz with a ±8 g dynamic range and 16 mg resolution using a special-purpose MEMS-accelerometer-based actigraphic device we developed [2]. To reduce deterministic measurement errors (scale, offset, and orthogonality), the acceleration signals were subjected to post-calibration using a publicly available algorithm [44].

#### 2.3.2. Similarity-Matrix-Based Comparison

The impact of actigraphic signal processing on sleep–wake scoring and each NPCRA indicator was examined through separate similarity matrices, in which each cell $M\left[i,j\right]$ quantifies the discrepancy across the 42 subjects that would arise in the given indicator value or sleep–wake score if the subjects had been effectively measured using an alternative actigraphic device—and thus their locomotor activity would have been quantified through a different, manufacturer-dependent method (e.g., MW vs. AC).

For NPCRA indicators with scalar outcomes (L5val, M10val, RA, IS, and IV), the Symmetric Mean Absolute Percentage Error (SMAPE) was used as a similarity measure to facilitate comparability with the correlation-based similarity matrix (Figure 2). The method of computing a single similarity-matrix cell $M\left[i,j\right]$ is defined in Equation (4), where N is the number of participants; for participant s, $V_{s}^{i}$ and $V_{s}^{j}$ are the values of a given indicator (e.g., L5val), derived from actigraphic signals produced by signal-processing pipelines i and j. SMAPE ranges from 0% to 200%, with lower values indicating smaller discrepancies in the given indicator that are attributable to the choice of signal-processing pipeline (e.g., device selection). Note that L5val and M10val are not ratio-type measures (unlike RA, IS, and IV); therefore, the differences in the value ranges of differently computed activity data might introduce order-of-magnitude discrepancies in these indicators [13,45]. Thus, for fair comparison of L5val and M10val, we have normalized the actigraphic data by their respective means to reduce such distortions.(4)$$M\left[i,j\right]=\frac{1}{N}\sum_{s=1}^{N}\frac{\left|V_{s}^{i}-V_{s}^{j}\right|}{\left(\left|V_{s}^{i}\right|+\left|V_{s}^{j}\right|\right)/2}\cdot 100$$

For NPCRA indicators for which the outcome of interest is the onset (L5onset and M10onset), the similarity matrices were constructed from the overlap (ranging from 0% to 100%) of the designated windows, as defined in Equation (5). Here, N denotes the number of participants; for participant s, $W_{s}^{i}$ and $W_{s}^{j}$ are the designated windows for the given indicator, derived from actigraphic signals produced by signal-processing pipelines i and j, and L is the segment length (5 h for L5onset and 10 h for M10onset).(5)$$M\left[i,j\right]=\frac{1}{N}\sum_{s=1}^{N}\frac{\left|W_{s}^{i}\cap W_{s}^{j}\right|}{L}\cdot 100$$

For sleep–wake scoring algorithms, two aspects have been evaluated: the timing of sleep and Total Sleep Time (TST). In the former case, analogous to onset-based NPCRA indicators, overlap was assessed; however, because the lengths of sleep segments were not uniform, overlap was quantified via the intersection over union (IoU), i.e., the Jaccard index, as in Equation (6). In the formula, N denotes the number of participants; for participant s, $S_{s}^{i}$ and $S_{s}^{j}$ are the designated sleep intervals from actigraphic signals, produced by signal-processing pipelines i and j.(6)$$M\left[i,j\right]=\frac{1}{N}\sum_{s=1}^{N}\frac{\left|S_{s}^{i}\cap S_{s}^{j}\right|}{\left|S_{s}^{i}\cup S_{s}^{j}\right|}\cdot 100$$

Additionally, we employed dendrograms to reveal the structure and hierarchical organization of the similarity matrices. Prior to this, matrices were converted to dissimilarities and rescaled between 0 and 1, with lower values indicating smaller differences. Dendrograms were computed using SciPy (v1.7.3)—with the default Euclidean metric and complete linkage—which is less sensitive to outliers and noise [46].

In this work, we also connect our prior findings about the correlation-coefficient-based relationships between the different activity determination methods (see [2] for details) to present results. Figure 2 displays the correlation-based similarity matrix, which not only aids in the interpretation of our current results (see Section 3) in light of our previous findings but also serves to illustrate the rationale behind our matrix-based comparison. In the correlation matrix, the closer the r value is to 1, the stronger the linear relationship between activity signals derived from the same acceleration data (see Section 2.3.1) but computed using different methods (Section 2.2.1). These relationships are mainly driven by how the acceleration signal was preprocessed, an observation we reinforced by color coding the labels to indicate different preprocessing families: purple (raw per-axis acceleration), light and dark blue (per-axis filtered acceleration), red (magnitude-filtered acceleration), and green (unfiltered magnitude of acceleration).

## 3. Results

In the following, we first examine the impact of the actigraphic signal processing methods encompassed in our generalized activity-determination framework on the values of NPCRA indicators (see Section 3.1.1) and then on their onset (see Section 3.1.2). This approach allows us to independently investigate the effects of the two main steps of activity determination: preprocessing of the acceleration signal and its compression into activity values using activity metrics. However, because linking the generalized methods to actual devices is not trivial, we also demonstrate how specific activity determination methods used with widely adopted instruments in the related literature influence NPCRA indicator values (see Section 3.2.1) and sleep–wake scoring (see Section 3.2.2).

### 3.1. Effect of Generalized Activity Determination Methods

To fully characterize the influence of signal-processing steps within our generalized activity-determination framework, we have applied NPCRA indicators not only to epoch-level activity data but also to the underlying, differently preprocessed higher-temporal-resolution acceleration data—an unprecedented approach for which we first had to explore the technical constraints. Since nonparametric measures (Section 2.1.2) rely on averaging, they fail for time series with constant means. For raw magnitude data (UFM), positive and negative accelerations cancel each other out, so taking its average over even a-few-second-wide windows converges to Earth’s gravitational constant (1 g). Removing the gravitational component with band-pass filtering late (FX, FY, FZ, and FMpost) yields signals that are near-symmetric around 0 g, whose mean will also be constant unless absolute values are taken. Consequently, NPCRA measures were only applicable to FMpre, HFMpre, UFNM, and ENMO without technical constraints. For FX, FY, FZ, and FMpost, the absolute value had to be taken. It should be noted that the 24 h mean profiles for acceleration data were obtained at a 1 s resolution rather than 1 min, reflecting their higher temporal resolution compared to epoch-level activity data.

#### 3.1.1. On the Value of NPCRA

Figure 3 illustrates how the mean values of L5, M10, and RA across 42 subjects depend on which signal-processing pipeline in our generalized framework was used to determine the acceleration or activity signal. For each NPCRA indicator, 43 mean values are represented with colored dots: 35 for those activity signals that previously formed the rows and columns in the correlation matrix of Figure 2, and 8 for acceleration signals that have undergone various preprocessing procedures. The coloring of the dots in Figure 3 indicates three categories: blue denotes values computed from epoch-level activity data derived with threshold-based metrics (TAT and ZCM), red denotes values from activity data derived with non-threshold metrics, and black denotes values computed directly from higher-temporal-resolution acceleration data preprocessed in different ways. As shown, applying the L5 indicator to threshold-based activity signals (blue) yields systematically lower values than applying it to activity signals derived with any non-threshold metric (red). By contrast, for M10, the pattern reverses: threshold-based metrics yield higher values. Notably, the indicator values computed from acceleration signals (black) closely resemble those computed from activity signals obtained with non-threshold metrics (red). Since RA is derived from L5 and M10 values, it also forms two distinct clusters, similar to those observed for the two aforementioned indicators.

In Figure 4, the counterpart to the previous figure is shown for the mean values of the IV and IS indicators. In this case, however, the color coding of the mean values—contrary to Figure 3—is based on the preprocessing families, since they group according to the preprocessing techniques applied to the acceleration signal and not based on whether the activity metric was threshold-based or not. These clusters that clearly resemble the structure of the correlation-based similarity matrix (see Figure 2) appear in the same order for IS and IV—albeit with opposite directionality.

Figure 3 and Figure 4 clearly demonstrate that the values of all the NPCRA indicators considered can differ substantially depending on how the actigraphic signals are derived within our generalized framework. For example, IV was reported to be around 0.67 for the controls and 0.811 for subjects suffering from bipolar disorder [47], while IS was measured at approximately 0.446 for the controls and 0.526 for schizophrenic subjects [48]; but as shown in Figure 4, even greater discrepancies can occur within the same group of healthy subjects due to the lack of standardization in actigraphy. Moreover, different NPCRA indicators are more strongly influenced by different aspects of the actigraphic methodology: for the values of L5, M10, and RA, the activity metric exerts the greater influence, whereas for IS and IV, the preprocessing technique applied to the acceleration signal has a larger effect. The impact of these aspects has been quantified using similarity matrices and dendrograms, as defined in Section 2.3.2.

Figure 5 presents the SMAPE-based similarity matrices and their corresponding dendrograms for L5 values; the labels follow the same categorical color coding as for Figure 3. It makes what was inferred from the means in Figure 3 even clearer: the L5 values differ substantially (the differences are generally greater than 80%) depending on whether L5 was applied to activity signals derived using threshold-based metrics (blue labels), to activity signals derived using other metrics (red labels), or directly to acceleration signals (black labels). The corresponding rationale will be addressed in Section 4. For the values of M10 and RA (corresponding matrices are provided in Appendix A, respectively), the same clustering pattern was observed. However, while the SMAPE-based difference between the two clusters was generally greater than 80% for L5, it is below 10% for M10 and around 20% for RA. Since RA is directly derived from the value of M10 and L5, it means that the observed discrepancies for RA are mainly driven by the differences in the value of L5.

As shown in Figure 6, for the IS indicator (the corresponding matrix for IV can be found in Appendix A), the differently determined acceleration and activity signal types form hierarchical clusters along the choice of acceleration-signal-preprocessing technique—mirroring the correlation-based similarity matrix of Figure 2 and the separation of mean values observed for IS and IV in Figure 4—but the differences between the clusters are generally smaller compared to what we observed for the value of L5 in Figure 5. An important implication follows: for a given preprocessing technique applied to the acceleration signal, IS and IV computed directly based on acceleration data generally agree closely with those computed from activity signals derived from the same acceleration data using any activity metrics. In contrast, the values of L5, M10, and RA can differ substantially depending on whether they are computed from acceleration signals or activity signals if the latter were derived using threshold-based metrics.

#### 3.1.2. On the Onset of NPCRA

Figure 7 illustrates the extent of overlap between the 5 least active and 10 most active 10 h periods identified by L5 and M10. As shown, both matrices exhibit similar structures; however, the onset of L5 appears less affected by the choice of actigraphic methodology, as the designated periods generally overlap to a much greater extent compared to M10. Additionally, the patterns of these matrices resemble those observed for IS and IV (see Figure 6) or in the correlation analysis (see Figure 2), suggesting that the onset of L5 and M10 is primarily influenced by how the acceleration data were preprocessed rather than the choice of activity metric used for epoch-level data compression, a finding that is the opposite of what we observed for the values of L5 and M10 (see Figure 5). This further reinforces the necessity of considering the entire activity-determination process when comparing results across studies, as L5 and M10 are largely influenced by both main steps of the actigraphic signal-processing pipeline.

### 3.2. Effect of Activity Determination of Specific Devices

To demonstrate how selecting from among specific actigraphic devices used in the literature affects the values of NPCRA indicators and sleep–wake scoring, we performed the analysis on a narrower set of activity determination methods of specific devices. These methods were applied to the raw acceleration data to emulate the outputs of those specific devices. As defined in Section 2.2.2, this narrower set includes the algorithms of AMI and Condor Instruments devices, namely, PIM(FMpre), ZCM(FMpre), and TAT(FMpre); PIM(ENMO) of the GGIR package from the above analysis; and, as new additions, the specific algorithms used in Ametris and CamNTech devices (AC and MW, respectively).

#### 3.2.1. On the Value of NPCRA

Figure 8 presents the spread of L5, M10, and RA indicator values (each on separate panels, (a) to (c)) across the 42 subjects for specific devices. Since AC and MW are also threshold-based activity metrics like ZCM and TAT, the values form clusters within each indicator depending on whether the activity data were calculated using threshold-based activity metric. This clustering is further substantiated by the dendrogram in panel (d), which is based on the RA. These findings are in line with the observations we made regarding the generalized activity determination methods (see Figure 3). For example, the device specificities of RA are mainly due to the differences in the value of L5, as the interquartile ranges of different devices overlap for the value of M10, but not for the value of L5 and RA. It is crucial to highlight that ZCM, TAT, and PIM are the three classical activity metrics supported by both AMI and Condor Instruments devices. However, since selecting between them is up to scientists when preparing for measurement, only this choice can have a substantial impact on the NPCRA outcomes as shown. For example, the RA value was, on average, close to 0.70 for PIM and 0.86 for ZCM (i.e., an average difference of 0.16 for an indicator that ranges between 0 and 1), and we found that their subject-level difference was statistically significant using a paired *t*-test (*p* < 0.001). Since RA values were derived for the same motion, these discrepancies are purely a consequence of how the actigraph was set up (or which device was used), and, remarkably, they are greater for 38 out of the 42 subjects compared to what was observed between pediatric subjects and adults by others using the same device [13].

Figure 9 shows the counterpart of the previous figure for the IS and IV indicator values in panels (a) and (b), while panel (c) depicts the dendrogram for IS. The dendrogram clearly shows that the IS derived from activity data based on the activity determination method from GGIR (i.e., PIM(ENMO)) substantially differs from that derived via the rest of the methods. Since PIM(ENMO) is the only activity determination method in this narrower set where digital filtering was not involved in the preprocessing of acceleration data, this behavior aligns with the observation we made regarding the generalized activity determination methods (see Figure 4): the differences device selection can introduce in the IS and IV are primarily driven by how the acceleration data have been preprocessed rather than the activity metric. For both IV and IS, the greatest discrepancies occurred between PIM(ENMO) from the GGIR package and ZCM(FMpre) supported by the AMI and Condor Instruments devices (for example, on average, 0.99 vs. 0.75 for IV, respectively).

#### 3.2.2. On Sleep–Wake Scoring

The limited set of specific devices’ activity determination methods did not allow us to comprehensively describe the similarity structure based on NPCRA indicator onset times as we did for the generalized framework, for which we compared 43 methods (Section 3.1.2) instead of only 6; therefore, we instead examined another timing-related metric in this case: sleep–wake scoring. We assessed the impact of device choice on sleep–wake scoring by classifying sleep and wakefulness from raw acceleration data using the van Hees algorithm and by applying the MASDA algorithm to six different types of activity signals computed from the same acceleration data to emulate specific devices commonly used in the related literature. In Figure 10, one can see, for an exemplary subject, that even inter-device selection between ZCM and PIM activity metrics in AMI and ActTrust devices can result in several hours of discrepancies in sleep–wake scoring.

To systematically assess such discrepancies, we first examined the total amount of sleep accumulated over 10 days, on average, among 42 participants, depending on whether sleep was identified from raw acceleration data using the van Hees algorithm or from activity signals using the MASDA algorithm; this comparison is illustrated in Figure 11a. For MASDA, highly similar total sleep durations were obtained across activity signals derived from threshold-based metrics (TAT(FMpre), ZCM(FMpre), AC, MW) ranging from 79.3 to 80.3 h. In contrast, activity signals based on integration yielded shorter sleep durations: PIM(FMpre) produced roughly 4 h less sleep on average than the threshold-based signals, while PIM(ENMO) showed an exceptionally low TST of only 50.6 h. As shown earlier (see Figure 9), the PIM(ENMO) signal also behaved anomalously for the IV and IS indicators, likely due to the absence of digital filtering during acceleration preprocessing. Sleep–wake detection from the raw acceleration signal yielded a mean total sleep duration comparable to that estimated by MASDA for the PIM(FMpre) signal.

Beyond the similarity in TST estimated via the different methods, it is also crucial to assess how consistently they identified the onset of sleep. To this end, we examined the overlap of sleep segments identified through the different approaches using the IoU metric, as shown in Figure 11b. Given that the MASDA algorithm produced different TSTs when applied to PIM(ENMO), it is evident that it also identified sleep at substantially different times compared to other methods, with overlaps below 62% in all cases. Comparing acceleration- and activity-based sleep–wake scoring highlights the importance of overlap analyses, since TST alone fails to capture the temporal agreement between methods. While sleep–wake scoring based on acceleration data (the van Hees algorithm applied to UFXYZ) yielded an almost identical TST to the MASDA algorithm applied to activity data, the IoU-based overlap of corresponding sleep segments was consistently below 78% in all cases, underscoring the methodological difference. For instance, although TST differed by only 0.67% between UFXYZ and PIM(FMpre), their sleep segment overlap reached only 74.8%; this disagreement is also illustrated in Figure 12 for one representative participant.

The parameters of the MASDA and van Hees sleep–wake scoring algorithms could be fine-tuned for each signal type to reduce their discrepancies, especially for MASDA when applied to PIM(FMpre) activity data. However, even tuning on specific datasets is not a common practice in typical use cases in the related literature, which we also aimed to replicate in order to examine the extent to which manufacturer-specific actigraphy signal processing affects sleep–wake scoring. Even the fact that we did not simply retain the default parameters, but instead selected MASDA settings within the recommended range that yielded acceptable results across the differently derived activity signals, already represents a more careful approach. Note that we used sleep–wake scoring solely to illustrate the discrepancies introduced by the actigraphic methodology of specific devices from a timing-oriented perspective (similar to the onset of L5 and M10 in Section 3.1.2) without focusing on sleep detection performance; therefore, we did not aim to evaluate the specificity or selectivity of the different sleep–wake scoring algorithms in the absence of a gold standard.

## 4. Conclusions and Discussion

Overall, it is evident that not only the NPCRA indicators but also the sleep–wake scoring algorithms evaluated are influenced by how the underlying movement data are quantified—in other words, by the type of actigraphic device used for data collection. For the IV and IS indicators, as well as the timing of the L5 and M10 indicators and sleep–wake classification, the primary source of difference was the initial step of the activity determination pipeline—that is, the preprocessing of the acceleration data, similar to what we previously observed by comparing activity signals through correlation—where both the fact that digital filtering was applied and the manner in which it was performed played central roles. Conversely, for the values of L5, M10, and RA indicators, the main differences were not due to preprocessing but rather to whether a threshold-based or non-threshold-based activity metric was used in the following step to compress the higher-temporal-resolution acceleration data into epoch-level activity values. Because the discrepancies we observed in the NPCRA analytical outcomes that are solely due to differences in actigraphic methodology may exceed those typically reported as being significant between subject groups with different health conditions in the literature, and as studies often analyze activity data recorded by actigraphs from various manufacturers operating under different principles, particular caution is required when comparing findings across studies. Unfortunately, this issue is further complicated by the fact that studies rarely provide sufficient detail on how their activity data were collected.

Figure 13 provides an example for L5 and M10, illustrating their dichotomy. On the one hand, when the activity metric is fixed but preprocessing differs (red vs. blue curves), average activity levels within windows are similar, though their onset varies. On the other hand, when the preprocessing technique is fixed but activity metrics differ in terms of whether they involve thresholding (black vs. blue curves), the timing of the windows aligns closely, but their mean activity differs—especially for L5. The reason for this is that L5 reflects nocturnal activity; during sleep, apart from infrequent rolling over in bed, the acceleration values recorded by the actigraph become so small that they fail to generate threshold crossings or the time above threshold. Consequently, the threshold-based ZCM and TAT activity metrics often produce epochs with zero activity, leading to lower L5 values. By contrast, acceleration values (after the gravitational component has been removed) remain nonzero, even if they are very close to zero. Therefore, other activity metrics, such as PIM, which integrates acceleration over time, yield small but nonzero activity values and hence higher mean activity. It is intriguing that preprocessing acceleration data primarily influenced the onset of these indicators rather than their values; however, understanding the underlying rationale requires further investigation. It should be emphasized that in our analysis, differences arising from the distinct value ranges of various activity signal types were mitigated by normalizing both acceleration and activity signals by their respective means before computing L5 and M10. However, this step is not a standard part of general NPCRA analyses, meaning that across studies, even-greater differences in these indicators may occur.

A further novelty of our work is the extension of NPCRA indicators beyond epoch-level activity data to higher-temporal-resolution acceleration signals, which we also incorporated into our comparative analyses. Through this, we have demonstrated that NPCRA indicators can be flexibly applied to acceleration data from which the gravitational component has already been eliminated, and NPCRA indicators derived from them showed high agreement with the ones derived from activity data calculated with non-threshold-based activity metrics. However, this on its own will not inherently enhance comparability across studies, since even when experts use actigraphs that record raw acceleration signals, it remains customary to compute activity retrospectively using publicly available methods (e.g., the Ametris solution [37]), which often involve employing threshold-based activity metrics—and thus we return to the fundamental issue. This approach stems from the fact that classical analytical methods (e.g., NPCRA) were developed when only devices capable of collecting activity signals were available. While manufacturer-specific activity determination methods undermine cross-study comparability, using activity data also eases practical constraints: compared with triaxial raw acceleration sampled at 100 Hz, 60 s epoch-based activity data contain 18,000 times fewer data points, substantially reducing storage requirements and computational demands during data analysis. Additionally, activity data computed using non-threshold-based metrics exhibit features like acceleration data—as demonstrated in the present study and in our previous work, where we showed their agreement in the frequency domain [27]. Despite these factors, we believe that collecting acceleration data offers advantages from a device-engineering standpoint (by reducing on-device computations and potentially extending battery life, albeit at the cost of increased on-board storage usage) as well as by enabling direct analysis of acceleration data itself, thereby at least avoiding the inhomogeneity of activity metrics, which represents a step in the right direction. However, the standardization of acceleration preprocessing techniques is still necessary.

Although we aimed to apply a general and inclusive approach for comparison, the emulation of device specificities necessitated a few unavoidable simplifications, which may serve as limitations of our results. These include, for example, the fact that post-calibration of acceleration data could have also been performed using a different algorithm (e.g., the built-in solution in GGIR). Given that the correlation pattern (Figure 2) required a fixed epoch length across activity determination methods, the ENMO acceleration data were compressed into 60 s epoch-level activity values like the others, even though the default epoch length in GGIR is 5 s. We specifically compared the NPCRA outcomes for ENMO between 5 s and 60 s epoch lengths, and we observed near-perfect alignment (the overlap of the L5/M10 windows was >99%, and the SMAPE-based differences for L5, M10, RA, IS, and IV were <0.1%); thus, our results were not affected by this choice. The filter characteristics specific to MotionWatch devices could only be approximated due to the lower sampling rate for the acceleration data analyzed. Although we could not directly examine its effect due to the same reason, the comparison of NPCRA outcomes for high-pass-filtered (HFMpre) and band-pass-filtered acceleration data (FMpre, FMpost) with unfiltered acceleration data (UFM, UFNM, and ENMO) indicated that the differences were not driven by the different filter characteristics themselves but by the time when the filter was applied to the acceleration data during preprocessing (before or after the magnitude calculation). This suggests that the filter approximation of MotionWatch devices did not introduce substantial bias in examining sleep–wake cycles. Additionally, since our objective was only to illustrate whether sleep–wake scoring algorithms are affected by the actigraphic methodology as a supplement to our NPCRA-based examinations, the absence of a PSG-based ground truth was not a limiting factor in our case. However, assessing the algorithms’ scoring performance in the context of manufacturer-dependent signal-processing pipelines for activity data represents an intriguing direction for further research. While acknowledging these technical constraints contributes to a more comprehensive interpretation of the findings, these factors either did not limit us or do not appear to have substantially affected the results presented (as also supported by our preliminary observations from ongoing analyses), and they certainly do not alter the primary conclusion that differences in manufacturer-specific signal-processing techniques employed in actigraphs can markedly influence the outcomes of circadian rhythm and sleep–wake analyses. By highlighting this, our findings may also aid medical-oriented research in distinguishing between healthy subjects and those with different health conditions [9,49].

## Figures and Tables

**Figure 1 sensors-25-07659-f001:**
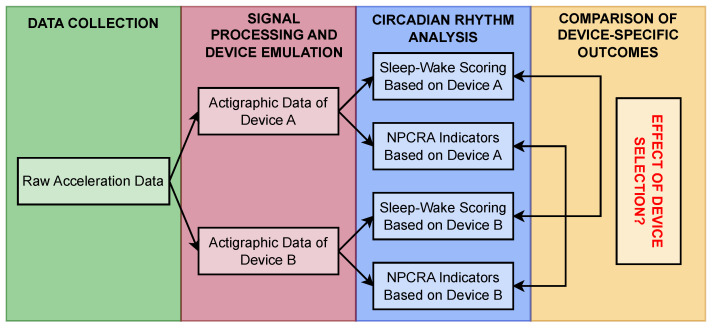
A simplified flowchart of our analytical approach.

**Figure 2 sensors-25-07659-f002:**
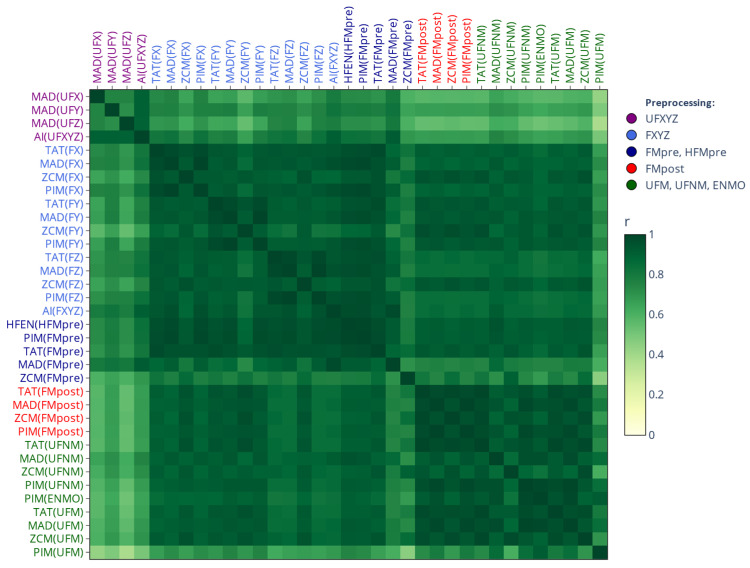
Correlation-coefficient-based similarity among different activity determination methods [2]. The cells indicate the average correlation, computed across 42 subjects, between activity signals derived from the same motion but determined differently according to our generalized framework. Row and column labels are grouped by the employed acceleration signal-preprocessing technique (which is in parentheses, as the argument of the activity metric) and color-coded accordingly.

**Figure 3 sensors-25-07659-f003:**
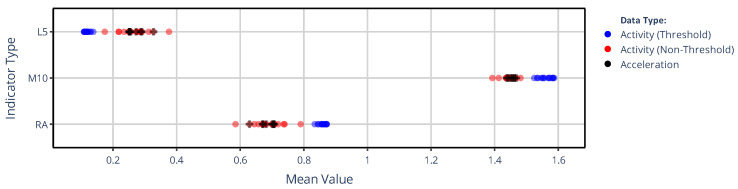
The mean values of L5, M10, and RA across 42 subjects are shown depending on which signal-processing pipeline of our generalized framework was used to generate the different acceleration and activity signals. The mean values are grouped into 3 categories by color coding: black circles denote mean values derived from acceleration data, while blue and red circles represent mean values for activity signals determined using threshold-based and non-threshold-based activity metrics, respectively.

**Figure 4 sensors-25-07659-f004:**
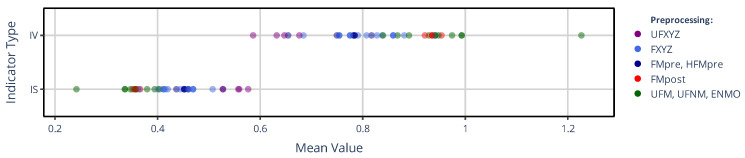
The mean values of IS and IV across 42 subjects are shown depending on which signal-processing pipeline of our generalized framework was used to generate the different acceleration and activity signals. In contrast to Figure 3, but similar to Figure 2, the mean values are grouped into 5 categories by color coding according to the preprocessing technique applied to the raw acceleration data, as indicated in the legend.

**Figure 5 sensors-25-07659-f005:**
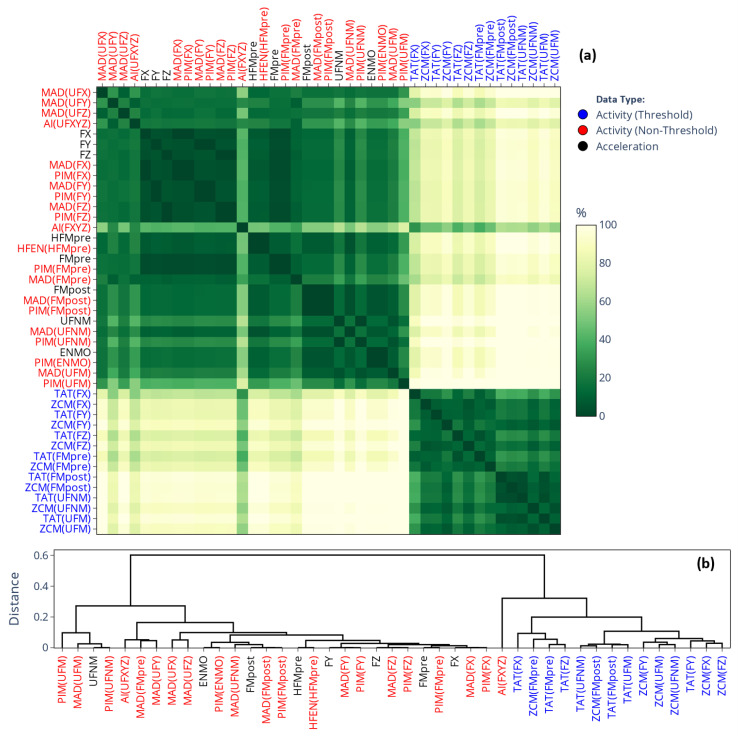
(**a**) Differences in the mean value of L5 across 42 subjects depending on which signal-processing pipeline of our generalized framework was used to generate the acceleration or activity data. The cells of the matrix represent SMAPE values ranging from 0% to 200%, with the gradient color scale shifting linearly from green to yellow as SMAPE increases from 0% to 100%. (**b**) The dendrogram corresponding to the similarity matrix. The label colors in both the similarity matrix and the dendrogram follow the same color coding as in Figure 3, as indicated in the legend.

**Figure 6 sensors-25-07659-f006:**
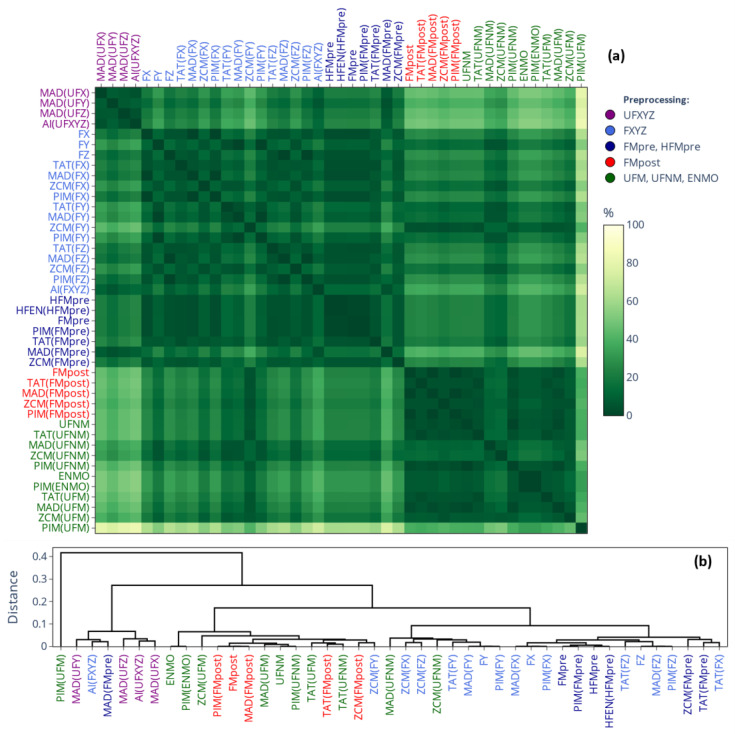
(**a**) Differences in the mean value of IS across 42 subjects depending on which signal-processing pipeline of our generalized framework was used to generate the acceleration or activity data. The cells of the matrix represent SMAPE values ranging from 0% to 200%, with the gradient color scale shifting linearly from green to yellow as SMAPE increases from 0% to 100%. (**b**) The dendrogram corresponding to the similarity matrix. The label colors in both the similarity matrix and the dendrogram follow the same color coding as in Figure 2 and Figure 4, as indicated in the legend.

**Figure 7 sensors-25-07659-f007:**
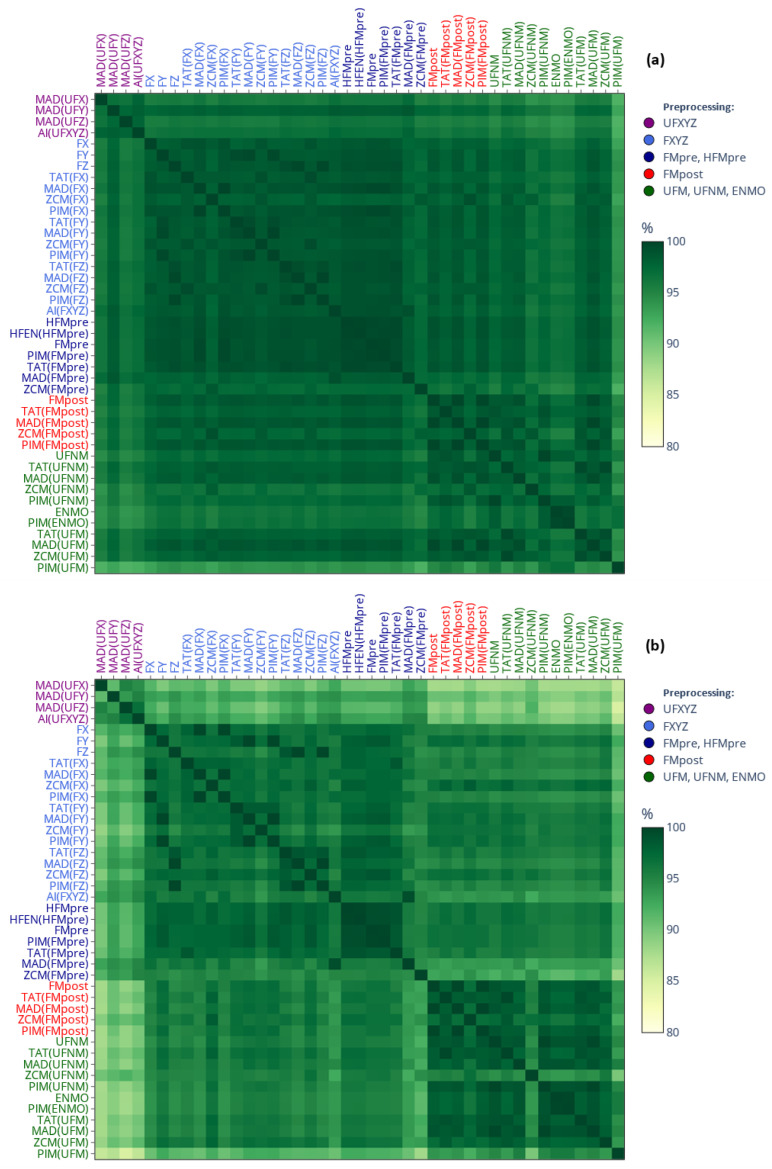
Differences in the onset of L5 (**a**) and M10 (**b**) across 42 subjects depending on which signal-processing pipeline of our generalized framework was used to generate the acceleration or activity data. The cells of the matrix represent the mean percentage overlaps between the designated windows ranging from 100% to 0%, with the gradient color scale shifting linearly from green to yellow as overlap decreases from 100% to 80%. The label colors in both the similarity matrix and the dendrogram follow the same color coding as in Figure 2 and Figure 4, as indicated in the legend.

**Figure 8 sensors-25-07659-f008:**
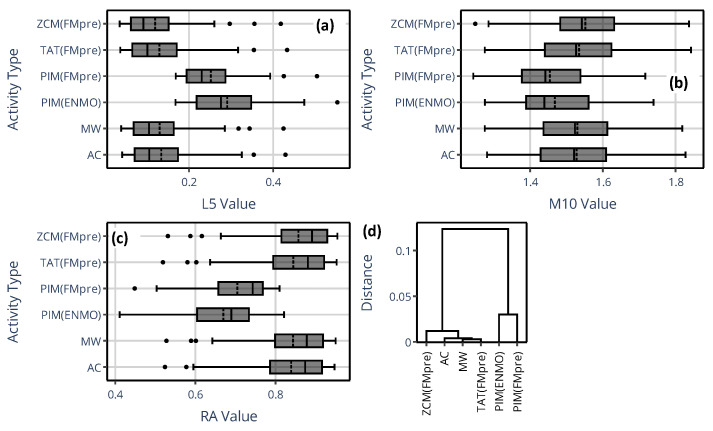
(**a**–**c**) The spread of L5, M10, and RA values across 42 subjects is shown using box plots depending on which specific actigraphic device was used to quantify locomotor activity. Between Q1 and Q3 quartiles (i.e., the interquartile range), the mean (dashed line) and median (solid line) are depicted. Values below Q1 minus 1.5 times the interquartile range, or above Q3 plus 1.5 times the interquartile range, were considered outliers and shown as black dots. (**d**) The dendrogram based on RA values.

**Figure 9 sensors-25-07659-f009:**
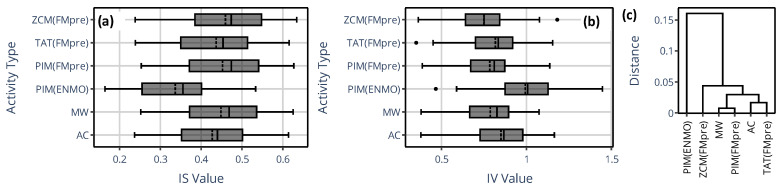
(**a**,**b**) The spread of IS and IV values across 42 subjects is shown using box plots depending on which specific actigraphic device was used to quantify locomotor activity. Between Q1 and Q3 quartiles (i.e., the interquartile range), the mean (dashed line) and median (solid line) are depicted. Values below Q1 minus 1.5 times the interquartile range, or above Q3 plus 1.5 times the interquartile range, were considered outliers and shown as black dots. (**c**) The dendrogram based on IS values.

**Figure 10 sensors-25-07659-f010:**
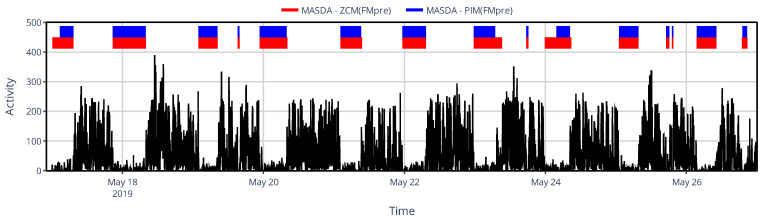
An example of sleep–wake scoring for a representative subject, showing designated sleep intervals obtained using the MASDA algorithm when the subject’s locomotor activity was quantified with AMI/ActTrust devices operated in ZCM (red) or PIM (blue) mode. The black trace shows the subject’s activity data to aid visual interpretation.

**Figure 11 sensors-25-07659-f011:**
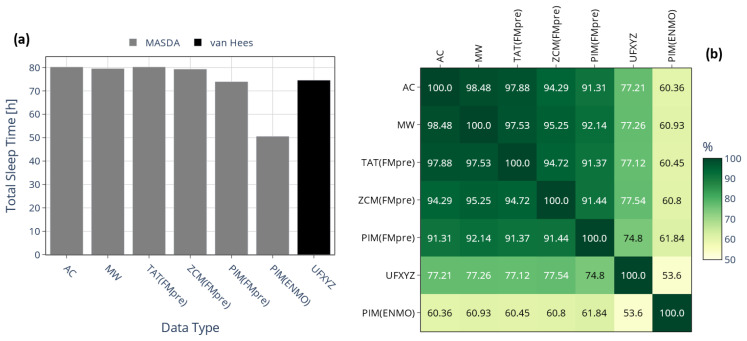
(**a**) The mean TST across 42 subjects is shown depending on whether sleep was scored based on the raw acceleration data using the van Hees algorithm (black bar) or by the MASDA algorithm based on the activity data of specific devices (grey bars). (**b**) The mean percentage overlaps between the aforementioned sleep segments, ranging from 100% to 0%, with the gradient color scale shifting linearly from green to yellow as overlap decreases from 100% to 50%.

**Figure 12 sensors-25-07659-f012:**
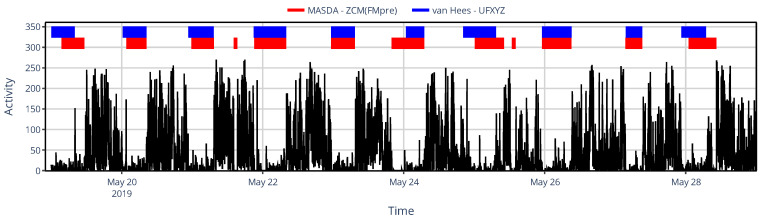
An example of sleep–wake scoring for a representative subject, showing designated sleep intervals obtained using the MASDA algorithm applied to the ZCM activity data of AMI/ActTrust devices (red) and the van Hees algorithm applied to the underlying, raw acceleration data (blue). The black trace shows the subject’s activity data to aid visual interpretation.

**Figure 13 sensors-25-07659-f013:**
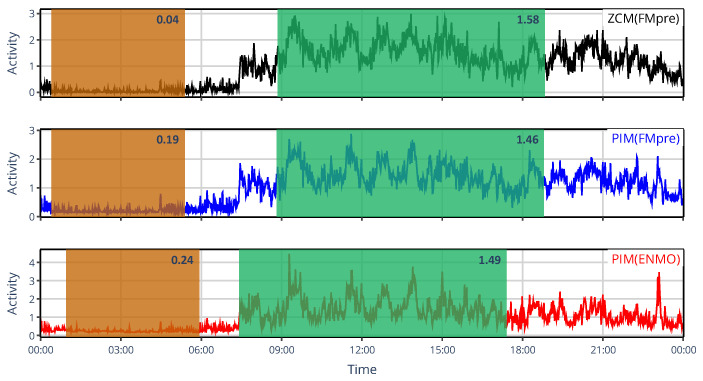
The 24 h mean activity profile of the same representative subject, whose locomotor activity was quantified using the ZMC(FMpre) (black), PIM(FMpre) (blue), and PIM(ENMO) (red) activity-determination methods. The designated L5 and M10 windows are shaded in orange and green, respectively, while the corresponding indicator values are shown at the top right corner.

## Data Availability

The raw actigraphic acceleration data examined in this study are openly available in FigShare at https://doi.org/10.6084/m9.figshare.16437684.v1.

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
