# Peer review of "The Impact of Quantifying Human Locomotor Activity on Examining Sleep–Wake Cycles"

_sensors, 2025, doi:10.3390/s25247659_

Round 1
Reviewer 1 Report
Comments and Suggestions for Authors
Manuscript examines the impact of various actigraphy signal processing methods on measures of motor activity and circadian rhythms. The authors propose a generalized comparison methodology, simulating various methods for processing raw accelerometer data and applying appropriate analytical approaches. The work stands out for its thorough analysis and the importance of the topic, given the popularity of actography in sleep research.
Today, sleep clinics, due to actigraphy low cost and ease of use, are actively implementing this method even for clinical purposes, such as objectively detecting sleep problems in patients.
This work, with its detailed description of all the issues involved in interpreting actigraphy results, is important for a wide range of specialists.
The results are undoubtedly relevant for the scientific community, which requires reliable, standardized methods for analyzing long actography records.
Among the comments, I would like to note the following:
1. A direct comparison of various actigraphy methods with sleep onset and offset assessment based on polysomnographic recordings would be very useful.
2. More detailed descriptions of the processing parameters and the algorithm should be included in Section 2.3.1, rather than just a reference to the group's previous works.
3. Figures 2, 3, 7, and 8 should include information on the spread of values ​​across the group of 42 study participants, i.e., present not only average values ​​but also, for example, box plots.
Figures 9 and 11 should be presented at an enlarged scale, i.e., "stretched" along the x-axis, for ease of perception.
4. The emulation of specific devices may, in some cases, distort the processing results, which may be worth mentioning in the conclusion.
Reviewer 2 Report
Comments and Suggestions for Authors
The authors use a single, high-resolution wrist-acceleration data set (42 healthy adults, 10 days, 10 Hz, ±8 g) to emulate 35 generic and 6 manufacturer-specific “activity-count” pipelines (filtering + epoch compression). They quantify how these pipelines alter five non-parametric circadian rhythm (NPCRA) indicators (L5, M10, RA, IS, IV) and two sleep-wake scoring algorithms (MASDA, van Hees). Similarity matrices (SMAPE, IoU, overlap) and dendrograms are used to visualise the divergence attributable to methodological choice rather than biology. The manuscript addresses an important reproducibility gap and provides a well-executed, transparent benchmarking resource. However, the absence of ground-truth sleep data, limited hardware diversity and incomplete statistical inference prevent it from being conclusive on “which method is better”. Thus, several points require clarification or improvement to enhance reproducibility and clinical relevance.
- The sleep-wake algorithms (MASDA, van Hees) were applied with fixed parameters. The authors note that tuning could reduce discrepancies but argue that this is uncommon in practice. A sensitivity analysis (e.g., testing parameter ranges) would strengthen the claim that observed differences are primarily due to signal processing rather than algorithm tuning.
- The approximation of device-specific filters (e.g., MotionWatch’s 3–11 Hz band-pass) using a 0.25–2.5 Hz filter may introduce bias. The authors should quantify the impact of this simplification or justify it with spectral analysis.
- For normalization of L5/M10 values, while normalization was applied to mitigate scale differences, its non-standard nature warrants discussion. How might the absence of normalization in typical NPCRA studies exacerbate discrepancies?
- Clarify the rationale for selecting 1-minute epochs for PIM(ENMO) despite GGIR’s default 5-second epochs. Justify this choice with a citation or empirical rationale.
- The study does not validate results against polysomnography (PSG) or another gold standard for sleep-wake scoring. While the focus is methodological, some discussion on how the discrepancies align with known PSG-actigraphy differences would strengthen clinical relevance.
Reviewer 3 Report
Comments and Suggestions for Authors
The study investigates the possible influence of signal processing techniques (filtering, compression, etc.) - namely methodological differences - on nonparametric circadian rhythm analysis (NPCRA) and sleep-wake scoring.
The paper briefly reports the main findings and their implications, thus providing the reader with an immediately understandable idea of the outcome achieved and the contribution given to the state-of-art. Remarkably, the authors have stated that the compression step (aggregation into epoch-level activity data) can be avoided in the actigraphy pipelines, which has a direct impact not only on the cross-manufacturer harmonization, but also and especially the computational efficiency of actigraphic devices, which may become more sustainable in the long term.
However, many points have to be addressed in order to enhance the quality. The main concerns the authors should carefully address are presented as follows.
Above all, the adopted style makes the reader lose focus and affects the paper reproducibility. This "presentation" issue can make any reader difficult to understand what the study has proposed compared to the state-of-art and what it has achieved.
Therefore, the manner in which the contents are arranged must be carefully and meticulously revised.
No statistical tests have been conducted, thus reducing the scientific soundness of any discussions derived from the outcomes.
Highlights
h.1) Consider splitting the first sentence into two sentences, one for the definition of actigraphic devices (compression of accelerometer data into activity data) and the other for the lack of standardization. This split has been well done in the Abstract in the sentences "Actigraphy quantifies [...] activity values" (lines 34-37).
h.2) The sentence "NPCRA indicators [...] these analytical outcomes the most" can be summarized.
Abstract
- The sentence "building on our previous work [...] through correlation analysis" can be removed to prevent the reader from losing focus.
1. Introduction
1.1) Before defining how actigraphy had been used in the past - i.e., based on compression - the usefulness of the proposed method should be clarified. For instance, in view of continuous monitoring in everyday settings, the authors should mention (even briefly) sustainability issues related to the conventional pipelines aggregating data and how the energetic consumption can be optimized, thus possibly increasing the battery lifetime.
1.2) The Introduction only cites one authors' previous work - though preparatory and pertinent - and lacks some paragraphs about related works to unequivocally and robustly highlight the novelty of the presented work. Consider summarizing the contents of the previous work to make space for a clear statement of research gaps.
1.3) A final paragraph about the paper organization must be added.
2. Materials and Methods
2.1) A preliminary brief description and a related depiction (e.g., https://doi.org/10.3390/s24072199, https://doi.org/10.1109/MeMeA65319.2025.11067994) of the proposed pipeline should be added before all the other contents of "Materials and Methods" Section. After that, the following structure should be followed for the sake of paper readability: "Dataset", "Actigraphy Pipeline" (with Subsubsections "Preprocessing", "Feature Extraction", "Performance Metrics", and "Statistical Analysis"), "Dataset", "Dataset", "Dataset", "Dataset", "Dataset".
2.2) This Section should focus only on what the authors have used (materials) and done (methods), without diving too deep into the comparison with other algorithms, which is more suitable for an additional Section "Related Works". Hence, the description of Sleep-Wake Scoring Algorithms can be moved from Subsection 2.1 to "Related Works, whereas only the sentences "since the acceleration data [...] 100 milliseconds apart" can be moved to Subsection 2.3 "Actigraphic Signal Processing".
2.3) All the contents of Subsection 2.2 can be moved into an additional Subsection describing the features extracted.
2.4) In Subsubsection 2.3.1, the authors should focus on a clear statement of what has been done, not what can be done, to process data in actigraphs.
2.5) The description of the datasets utilized in this work should appear before all the Subsections about processing choices. Hence, the introductory text of Subsection 2.4 and the text of Subsubsection 2.4.1. can be merged into one single section Acceleration Data.
2.6) Data have been normalized when computing the similarity matrix. However, most of works exploiting inertial sensors normalize acceleration data before extracting the indicators and segment the portions of interest in which the motor action is performed (https://doi.org/10.3390/s24072199, https://doi.org/10.1109/MeMeA65319.2025.11067994). Hence, computing similarity after segmenting the whole data and uniforming them in the range [0,1] is suggested.
2.7) No statistical tests have been conducted, thus reducing the scientific soundness of any discussions derived from the outcomes.
3. Results
- Why not calculate the computational time that results from a pipeline with and without data compression? This would directly provide the reader with quantitative hints on the higher sustainability of the proposed approach.
4. Discussion
- The paper could benefit from exploiting the outcomes to support the addition of statements, even speculative, that avoiding data aggregation may save computational time and make the actigraphy framework more sustainable.
- One "Conclusions" Section should be added to provide the reader with a brief insight into the main paper contribution, its main outcomes, and its future works. Possible future developments may regard the use of the proposed features in a classification pipeline to compare healthy and pathological behaviors, since the correlation-coefficient-based similarity among IMU channels has already been employed to optimize pathological gait recognition (https://doi.org/10.1109/MeMeA65319.2025.11067994 ).
Minor comments
m.1) Figures and tables must be centered (when possible) with respect to the text (e.g., Figure 1, Table 2, Figure 2).
m.2) Since the Figures are low-quality and grainy, they must be enhanced (consider increasing dpi at least to 600).
Grammar
g.1) The present perfect should be used for verbs describing what has been done by the authors in the presented work, whereas the past simple can be used when citing related works - e.g., "in this work, we have examined" instead of "we examined".
g.2) The sentences based on "not only [...], but also" should use one comma before "but also".
Round 2
Reviewer 2 Report
Comments and Suggestions for Authors
Authors answered my concerns. Suggestion is that it's better to cite more latest references, moreover, the reference 4 in the response letter wasn't added into the reference list of the revised manuscript.
Reviewer 3 Report
Comments and Suggestions for Authors
The authors have revised the manuscript by matching most of the previously highlighted points. For instance:
h.1) The first sentence has been split into two sentences, one for the definition of actigraphic devices (compression of accelerometer data into activity data) and the other for the lack of standardization.
h.2) The sentence "NPCRA indicators [...] these analytical outcomes the most" has been summarized.
- The sentence "building on our previous work [...] through correlation analysis" has been removed to prevent the reader from losing focus.
1.1) The main contribution was already clear in the previous version of the manuscript. The suggestion of mentioning sustainability issues related to the conventional pipelines aggregating data and how the energetic consumption can be optimized was oriented to make the Introduction and the whole paper more attractive, thus reaching a wider public of readers - the same is for the additional result of computational time for supporting statements about sustainability. Therefore, even some brief speculations on such matters are welcome, as authors have successfully added in the Discussions.
1.3) The final paragraph about the paper organization has been added.
2.6) The order of operations for normalization and L5/M10 computation has been well clarified in the manuscript to prevent the reader from misunderstanding this design choice, which may help mitigate the inter-device differences indeed.
m.1, m.2) Figures and tables are now centered (when possible) with respect to the text and high-quality in terms of dpi.
g.1) The grammatical issues about the usage of the present perfect and past simple tenses, as well as the "not only [...], but also" sentences have been corrected.
However, some points still have to be addressed in order to perfect the quality. The main concerns the authors should carefully address are split into major and minor comments, as follows.
1. Introduction
1.2) The authors have clarified that their previous work has been cited as the only one related work examining the effects of device selection on sleep-wake scoring. Since the earlier wording has led to underestimating the work novelty, such a clear statement as "only one work has previously investigated [...]" is suggested to avoid the same misunderstanding.
2. Materials and Methods
2.1-2.5) The preliminary brief description could have been inserted directly at the beginning of Section 2, as it would introduce what the authors have used (materials) and done (methods). Nevertheless, the integration is welcome and appreciable.
On the other hand, although the rationale behind the choice of the previous item description is much more understandable, the current version can still make it difficult for the reader to immediately comprehend all the methodological aspects. Hence, consider a slight rephrasing such that the new wording can unequivocally highlight these aspects. This was the original intention of the suggested structure - i.e., "Dataset", "Actigraphy Pipeline" (with Subsubsections "Preprocessing", "Feature Extraction", "Performance Metrics", and "Statistical Analysis") - rather than an unrecognizable presentation of the work.
Furthermore, a kind apology for the multiple repetition of the "Dataset" word is needed to be addressed.
2.7) The addition of boxplots is respectable. Nevertheless, statistical tests have been performed, but the resulting outcomes seem not to be included, unless any comparisons proved to be non-significant. Furthermore, for the sake of a higher readability, such tests should have been described in the dedicated Subsubsection "Statistical Analysis" of Section 2. Were post-hoc correction tests applied to the preliminary result of the paired t-test?
- The conclusions are no longer hidden within the Discussion section. However, possible future developments regarding the use of the proposed features in a classification pipeline to compare healthy and pathological behaviors can be added, since the correlation-coefficient-based similarity among IMU channels has already been employed to optimize pathological gait recognition (https://doi.org/10.1109/MeMeA65319.2025.11067994 ).
